# The effects of luck perception on consumer variety-seeking

Jianbin Zhao[1], Zheng Li[2]*

1 School of Economics and Management, East China University of Technology, Jiangxi, P.R. China,
2 College of Economy and Trade, Guangdong-Hong Kong-Macao Greater Bay Area Agricultural Product Circulation Research Center, ZhongKai University of Agriculture and Engineering, Guangzhou, Guangdong, P.R. China

* zhengli@zhku.edu.cn

## Abstract

Companies often use luck, which is thought to be a psychological force, as a marketing tool to perform marketing activities. This paper designed three experiments to verify the effects of luck perception on variety-seeking. Experiment 1 found that luck perception allows consumers to have more variety-seeking. Experiment 2 examined the mediating role of novelty-seeking motivation and found that luck perception improves the novelty-seeking motivation and makes consumers prefer variety-seeking. Experiment 3 examined the moderating role of need for cognitive closure. For consumers who have low need for cognitive closure, whether prime luck perception or not, their variety-seeking are not significantly different, while for consumers with high need for cognitive closure who are after luck perception primed, they have more variety-seeking. This study provides theoretical support for understanding luck scientifically and enlightens and guides companies to expand product lines, present new brands, and implement precision marketing.

## Introduction

Many people believe that using certain objects or rituals affects their or others' luck. They carry lucky charms, lucky objects, etc. and choose luck-related behaviors daily. Hence, luck seems to be an important source of power in their lives. For example, Jordan, the famous professional basketball player, chose to wear jerseys with lucky numbers after a bout of low performance, expecting to perform better on the court [1]. Darke and Freedmanfound that many well-educated, intelligent, and scientifically-minded scientists also carry amulets, expecting these lucky items to bring good luck to their experiments [2]. A scientist even claimed to eat Nestle's wafer chocolates and drink Coke Light on the morning of an experiment, hoping that they would help the experiment succeed.

Similarly, companies started using luck in their marketing, such as ticket bonus, hoping that some luck stimuli will change consumers' perception of luck and influence

**Data availability statement:** All relevant data are within the manuscript and its Supporting information files.

**Funding:** This study was supported by the National Natural Science Foundation of China (71962001); Humanities and Social Sciences Research Foundation of Ministry of Education (21YJC630069); Guangdong superior agricultural products foreign trade innovation team (2020WCXTD013); Study on ways and countermeasures to enhance competitiveness of aquatic products in Guangdong Province (2022ZDJS018); Jiangxi Province Talent Project (sg2023-21).

**Competing interests:** The authors have declared that no competing interests exist.

consumers' consumption behavior [3,4]. While firms use luck as a tool for marketing practices, there is a dearth of theoretical research on luck in marketing. The existing research has mainly focused on the gaming industry. For example, individuals who feel lucky become more optimistic and confident in irrelevant or uncontrollable scenarios, possess a stronger sense of control, are prone to overestimate the probability of winning when purchasing lottery tickets, and prefer to place heavy bets [2,5–7]. Indeed, in other consumer domains, consumers' purchasing behavior is often exposed to consumer risk due to information asymmetry, such as consumer variety-seeking [8].

Variety-seeking has the attributes of excitement and risk, and is the behavior of individuals who seek unfamiliar and exciting things because they are acted upon by the need for variety [9]. Therefore, in order to satisfy the need for excitement and risk-taking, consumers indulge in variety-seeking [8]. In this paper, variety-seeking is viewed as decision-making behavior under information uncertainty. According to the psychological power theory of luck, luck can enhance consumers' risk preferences. If consumers are stimulated by luck, their novelty-seeking motivation becomes strong, and they will prefer behaviors or products with exciting, risky experiences and be willing to try new or different products or services [6,10]. Novelty-seeking motivation is a prerequisite for consumers' thrill- or risk-seeking behaviors [11], so novelty-seeking motivation is the mechanism by which the sense of luck acts on variety-seeking. Additionally, Wronska et al. argued that need for cognitive closure, which is a cognitive motivation for consumers to eagerly seek answers during information processing, affects consumers' sensitivity to luck information, which subsequently influences the affective effects of luck stimuli [12]. Therefore, consumers' need for cognitive closure moderate the effect of luck on variety-seeking. The analysis above is only based on phenomenal observation and logical speculation, and the specific internal mechanism and boundary conditions are yet to be empirically tested.

In this paper, the influence of luck on variety-seeking was explored in depth through experimental research methods. The main research questions include the direct influence of luck on variety-seeking, the mechanism of novelty-seeking motivation in the effect of luck on variety-seeking, and the boundary conditions need for cognitive closure. The findings offer innovative insights into the influence of luck in the field of consumption, broadening the scope of research on luck effects in product or brand decision-making. Furthermore, they delve into the psychological effects of luck, providing valuable theoretical support for people to gain a scientific understanding of luck. From a practical viewpoint, this paper's findings will shed light on the expansion of product lines and the launch of new brands and provide guidance for enterprises to implement precision marketing as well as luck marketing.

## Literature review and hypotheses development

### Luck

Luck, generally understood as good luck, refers to the knowledge that in an unlikely, uncalculated, and uncontrollable situation, the outcome of the event is exactly as expected [2]. Luck in psychology was first studied pertaining to the attribution theory.

Based on the logical reasoning of causality, psychologists believe that there are three reasons for behavioral events: ability, effort, and luck, understood as a random contingency that affects social events extrinsically, erratically, and uncontrollably [13]. However, Darke & Freedman found that many people cannot accept that luck is random, and they always feel that luck accompanies them [2]. A more concise version is as follows: Because luck always seems to favor certain people (i.e., consistently lucky), researchers have begun to study luck in terms of individual differences, arguing that luck beliefs are traits that are intrinsic to the psychological and behavioral stability of individuals [13]. Research has shown that individuals' luck beliefs contain four types: general luck beliefs, rejected luck beliefs, personal good luck beliefs, and personal bad luck beliefs [14]. However, Thompson & Prendergast empirically concluded that luck beliefs consist of two types: belief in luck and personal luck [15].

Based on researchers' views of luck as a trait, luck is a psychological variable. Researchers began to study stateful luck, i.e., stimulation using luck-related words, objects, lucky events, or lucky numbers, which can temporarily change individual's luck perception. For example, Darke and Freedman allowed participants to experience lucky events, which could significantly change their luck perception and boost their self-confidence [2]. Jiang et al. argued that since luck is a trait of individuals, luck perceptions are stored in individual's deep memory [16]. Therefore, psychological priming can be used to stimulate luck perception, resulting in transient luck. Stateful luck research suggests that individuals stimulated by luck are prone to change their active self-concept, and such transient changes in self-concept can strongly influence behavior independently of stable and long-term luck beliefs [17].

The experience of luck triggers positive emotions in individuals, creates the illusion of control, and gives them more hope, confidence, and a desire to succeed [18]. Research has shown that individuals who feel lucky prefer lucky products [5], like to maintain a positive self-image, prefer risky decision-making, and tend to overestimate the probability of winning when purchasing a lottery ticket [16], correctly identify a greater number of words in a crossword puzzle task, perform better in memory games, and persevere on a task for a longer period of time [6,19,20].

In summary, luck can be either a trait variable (an individual's beliefs about luck) or a state variable (a transient form of luck generated by an individual). Research has shown that both idiosyncratic luck beliefs and stateful luck impact individual's risk preferences. However, given the practical applications of firms, stateful luck is more likely to be a tool for firms' marketing manipulation. Therefore, this paper investigates the effect of stateful luck on consumer variety-seeking.

**Variety-seeking**

Variety-seeking is the behavior of consumers who choose to try a new, different product or service or switch between old and new products, brands, or services, although they can choose their preferred product or service repeatedly [21,22]. Variety-seeking tends to exhibit consumption uncertainty and ambiguity, with a large degree of missing information, and can be viewed as decision-making behavior in the presence of information uncertainty [8,23].

Traditional theories consider variety-seeking as purposeless or random behavior. However, researchers currently believe that consumers variety-seeking purchase decisions can be influenced by marketers' manipulation. For example, consumer environment, individual, and product characteristics can influence consumer variety-seeking [24–26]. Recent research suggests that variety-seeking may also be influenced by factors beyond consumer awareness. Maimaran and Wheeler found that if consumers are exposed to a heterogeneous (or homogeneous) array of shapes, they are more likely to make variety-seeking choices in subsequent irrelevant decisions [27]. Additionally, consumers in narrow (as opposed to wide) aisles engage in more variety-seeking because variety behaviors allow space-constrained consumers to freely express their claims [28]. Fishbach et al. showed that consumers who prime boredom (as opposed to loyalty) engage in more variety-seeking, because boredom activates the desire to satisfy prior options [21]. Based on the metaphor "variety is the spice of life," Mukherjee et al. verified that spicy flavored metaphors stimulate consumers' desire for interesting things, giving them more variety-seeking tendencies in subsequent irrelevant choices [29].

The existing literature has revealed that objective environmental factors and individual and product characteristics as well as some metaphorical concepts affect variety-seeking. Therefore, this paper suggests that priming luck perception will also affect consumer variety-seeking.

## Luck perception, novelty-seeking motivation and variety-seeking

People use the essence of luck, as an external tool, to eliminate uncertainty in the environment. This external tool is luck that makes individuals feel that they are with powerful people [6]. According to the psychological power theory of luck, luck can be perceived as an extended power of the individual that can influence events in their favor [30]. Thus, luck becomes a power resource for consumers, allowing them to develop a positive cognitive bias that can make them overestimate their self-control in event processing and prefer risky decisions. For example, Darke and Freedman found that individuals who held luck beliefs purchased more lottery tickets after experiencing a luck event, but individuals who did not hold luck beliefs purchased fewer lottery tickets after experiencing a luck event [2]. Zhou et al. used a football lottery and found that luck had a positive effect on the frequency of lottery ticket purchases, with lottery players who believed in luck purchasing more tickets compared to those who believed in ability [31]. Lim and Rogers also found that individuals who felt lucky purchased lottery tickets more frequently but also made more variety-seeking choices [32].

Consequently, consumers who feel lucky believe that they have a supportive power and can use this power to control the outcome of events as well as their personal fate. Thus, they hold a positive mindset regarding various life events and become more accepting of unfamiliar products [8,22]. Therefore, consumers who feel lucky tend to exhibit more variety-seeking.

**Hypothesis 1**: Luck perception positively influences consumer variety-seeking.

Research on luck suggests that consumers who feel lucky become more energetic and possess more preferences for unfamiliar or novel choices. For example, consumers who feel lucky are more likely to use online banking, rather than a physical bank, to deposit money [16]. In financial markets, consumers who feel lucky have a higher interest in sudden and unfamiliar market information; they are more willing to pay for random shock signals in the lottery market [33]. It is evident that luck motivates consumers and develops their interest in and desire for novel and unknown things. Novelty seeking is the innate tendency of consumers to show pleasure or excitement for novel stimuli [34].

Based on the above analysis, we argue that consumers who feel lucky possess a strong motivation to pursue novelty.

The motivation for novelty seeking is one of the important factors influencing individual behavior. It represents consumers' innate tendency to exhibit pleasure or excitement towards novel stimuli, and serves as one of the driving forces for consumers to actively pursue new experiences [35]. This is manifested in two aspects: intrinsic novelty seeking is the individual's desire for novel and exciting stimuli; extrinsic novelty seeking is manifested in the individual's actual behavior to obtain new stimuli. Therefore, we believe that the mechanism of luck's influence on variety-seeking emerges from consumer novelty-seeking motivation. This is because variety-seeking is an expression of consumers' search for novelty and pursuit of maximizing psychological utility, which can satisfy consumers' novelty needs [27]. If consumers repeatedly choose a product, their stimulation for the product becomes weaker and fails to satisfy their novelty needs. on the other hand, found that when consumers have strong novelty-seeking motivation, they will change their routine purchasing behaviors, such as choosing new and unfamiliar products or brands. Meanwhile, when their novelty-seeking motivation is weaker, they will continue to maintain routine buying behavior [36].

In sum, luck perception can give consumers a power resource that enhances their novelty-seeking motivation, and variety-seeking can satisfy their novelty-seeking needs. Therefore, luck affects consumer variety-seeking, and novelty-seeking motivation is the mediating mechanism.

**Hypothesis 2**: Novelty-seeking motivation mediates the influence of luck perception on consumer variety-seeking.

## Moderating role of need for cognitive closure

When a concept is primed, recency effects have a stronger influence than frequency effects for a short period [37]. However, if consumers' behavioral decision-making tendency relies on long-term, stable perceptions, they are susceptible to the frequency effects, and short-lived luck will not have a strong impact on subsequent consumer behavior [12]. Therefore, after priming consumers' luck, their subsequent consumption behavior will be influenced by long-term, stable cognition.

When consumers choose variety-seeking, the decision-making information they obtain is often incomplete, and they need to gather new information. At this time, consumers' cognitive characteristics largely determine their information gathering and processing process as well as their decision-making behavior [12].

Need for cognitive closure, as motivational factors influencing individuals' ambiguous decision-making in uncertain situations, are composed of dimensions such as cognitive structuring, ambiguity aversion, decision decisiveness, expectation predictability, and psychological closure [38]. need for cognitive closure is both personality traits and context-dependent, and can be aroused by time pressure, psychological fatigue, environmental noise, and task attractiveness. Individuals with higher need for cognitive closure often follow the principles of decision immediacy and decision continuity, and have a strong tendency to grasp and freeze [39]. need for cognitive closure affect individuals' attitudes towards ambiguity and information processing methods. Those with higher need for cognitive closure have lower ambiguity tolerance and are more prone to heuristic information processing; whereas consumers with lower need for cognitive closure have higher ambiguity tolerance and are more likely to engage in analytical information processing. Individuals with high need for cognitive closure rely more on conventions or stereotypes during information processing, less on information search, and are prone to forming larger decision biases; whereas those with low need for cognitive closure have higher ambiguity tolerance during information processing, rely more on information search, and therefore make more accurate decisions. These research conclusions have been widely tested in the fields of social judgment, team interaction, consumer behavior, negotiation, and politics.

Research has shown that relative to consumers with low need for cognitive closure, consumers with high need for cognitive closure are cognitively lazy, tend to adopt heuristic, simplified and top-down information processing, are prone to make decisions based on some marginal or peripheral cues, and their behaviors are susceptible to proximate cause effects [12]. Liu et al. argued that in an uncertain environment, in order to form conclusions quickly, consumers with high need for cognitive closure rely on simple, unattached rules, and take recent events as the basis for decision-making [40]. Thus, recent event cues have a significant effect on the behavioral decisions of consumers with high need for cognitive closure. For consumers with a high need for cognitive closure, if they encounter cues of luck, they are easily stimulated by these cues, which means they are prone to experiencing a sense of luck. Consequently, they tend to use this perceived luck as a guide for variety-seeking choices and are inclined to engage in variety-seeking behaviors. Therefore, for consumers with a high need for cognitive closure, luck perception has a significant impact on their preference for variety-seeking.

On the other hand, consumers with low need for cognitive closure are motivated to avoid cognitive closure and are easily affected by potential cues (such as emotions and stable psychological cognition). They seek alternative cues as the basis for decision-making. Therefore, consumers with low need for cognitive closure are cognitively hardworking, and recency effect will not have a strong influence on their behavioral decision making [36]. When faced with variety-seeking choices consumption scenarios, consumers with low need for cognitive closure rely on central, stable mental cues to make decisions, and their variety-seeking preferences do not change significantly, whether they are stimulated by luck or not. Therefore, for consumers with low need for cognitive closure, the influence of luck perception on the preference for variety-seeking does not undergo significant changes.

In summary, we believe that consumers with different levels of need for cognitive closure possess different recency cause effects. For consumers with high need for cognitive closure, when making variety-seeking choices, they are

susceptible to pre-decision luck stimuli or experiences due to the strong influence of recency effects on behavioral decisions. For consumers with low need for cognitive closure, when making variety-seeking choices, the pre-decision luck stimulus or experience will not have a strong influence on their variety-seeking due to the weak influence of recency effects on behavioral decision making.

**Hypothesis 3**: The effect of luck perception on variety-seeking is only signifcant among the consumers with high need for cognitive closure, but insignifcant for consumers with low need for cognitive closure.

## Experiment 1 luck perception and variety-seeking

This experiment focuses on verifying the direct effect of luck perception on consumer variety-seeking. The experiment used subliminal priming to prime participants' luck perception.

### Methods

The research was approved by the Medical Service Center of East China University of Technology (ethics approval number 24019). 19/11/2022~19/11/2022, a total of 265 undergraduate business students participated in the experiment, with a mean age of 20.27 years (SD = 0.96); 118 (44.52%) were male. The subjects are required to have independent consumption experience. The experiment used a single factor analysis method (type of priming: lucky vs. unlucky vs. control), and participants were randomly assigned to the lucky (63), unlucky (75), and control (127) groups.

Participants' luck perception was first primed. A thematic story was used to prime participants' luck perception [16]. They were first shown a picture of a suit, told that it was the latest suit from a local high-end clothing company, and subsequently shown the price of the suit. In the lucky group, the price of the suit was displayed as "RMB 688" (8 is the same as the pronunciation of "enrichment" in Chinese, suggesting good luck). In the unlucky group, the price of the suit was displayed as "RMB 744" (4 is the same as the pronunciation of "die" in Chinese, which implies bad luck). In the control group, the suit prices were all neutral numbers. However, to correspond with the two previous experimental conditions, we designed two control groups. Additionally, the prices given in the experiment could cause an anchoring effect on the participants [16]. Therefore, to ensure that the price of the control group was consistent with that of the (un) lucky group, the control group was shown two prices, which had the same mean value as the price of the (un) lucky group. Specifically, for the control group corresponding to the lucky group, the displayed prices were "¥723" and "¥653" respectively. For the control group corresponding to the lucky group, the displayed prices were "¥767" and "¥721" respectively. To effectively prime the participants' luck after they had looked at the prices, they were asked to judge whether the cost of the suit was higher or lower than their viewed price.

Afterwards, a shopping scenario was set up. To minimize participants' ability to guess the experiment's purpose, some questions unrelated to consumption choices were included in the shopping scenario questions. Participants were told, "Tomorrow night, your dormitory is going to hold a party with the dormitory next door, you need to buy some drinks and snacks, and your task is to buy 10 cans of chips. Suppose that when you buy chips at the supermarket, you find that there are 4 brands and 12 types of chips to choose from, all of which have differ10nt flavors (Table 1)." Participants were asked to choose which brands and varieties of chips they would purchase and determine the number of each variety they wanted to purchase.

**Table 1. Varieties and brands of chips.**

| Brands | Varieties | | |
|---|---|---|---|
| Oishi | Sweet & Spicy Flavor | Coriander Flavor | Garlic Flavor |
| Lay's | Cucumber Flavor | Blueberry Flavor | Lime Flavor |
| Copico | Seaweed Flavor | Thai Chicken Flavor | Teriyaki Pork Cutlet Flavor |
| Pringles | Barbecue Flavor | Curry Crab Flavor | Crispy Chicken Flavor |

Finally, participants were asked to complete a research task on personality assessment. They were asked to answer several personality questions unrelated to the current experiment, with a question about their subjective perception of luck inserted in the middle of these questions to measure the priming effect of the preceding luck, and the extent to which they agreed that "I had very lucky today" on a 7-point Likert scale, where 1 = "Strongly Disagree" and 7 = "Strongly Agree."

## Results

Luck perception. The analysis of variance of participants' perceived luck across priming types showed that the lucky group rated their luck higher than the unlucky group [$M_{Lucky}$ = 4.65, SD = 1.12 vs. $M_{unlucky}$ = 3.28, SD = 0.91; $F(1, 136) = 62.75$, $P < 0.001$]. There were no differences in participants' reported luck perception in the correspondence control condition [$M_{lucky\ control\ group}$ = 3.50 vs. $M_{unlucky\ control\ group}$ = 3.42; $F(1, 125) = 0.16$, $P > 0.05$]. There was a significant difference in luck perception between the lucky group and the corresponding control group [$F(1, 113) = 40.77$, $P < 0.001$]. There was no significant difference in luck perception between the unlucky group and the corresponding control group [$F(1, 148) = 1.53$, $P > 0.05$]. These results suggest that the luck perception manipulation in this experiment was successful. Additionally, there was no significant difference in participants' familiarity with chips across priming types [$F(3, 261) = 1.44$, $P > 0.05$].

Variety-seeking can be understood as a function of the number of different brands and flavors in a consumer's purchase history. Consumers can seek variety across both brands and varieties within the same brand. There are four dependent variables in this experiment to represent the participants' choice of variety-seeking. The first dependent variable is the number of varieties chosen by the participants, which indicates that they seek variety of product. The second is the number of brands chosen by the participants, which indicates that they seek variety of product brands. The third dependent variable is the ratio of purchases of favorite varieties, that is, the ratio of purchases of participants' favorite varieties to the total purchases (the total number of purchases in the present experiment was set at 10). The larger the ratio, the stronger the participants' purchase of favorite product varieties and the weaker the preference for product variety. The fourth dependent variable is the purchase ratio of favorite brands, that is, the ratio of the participants' purchase of favorite brands to the total purchases (the total number of purchases set up in this experiment is 10). The larger the ratio, the stronger the participants' purchase of favorite brands and the weaker the preference for brand variety.

*Number of varieties selected.* The results showed that the number of varieties selected in the lucky group was greater than those selected in the unlucky group [$M_{lucky}$ = 6.40, SD = 1.43 vs. $M_{unlucky}$ = 4.51, SD = 0.94; $F(1, 136) = 86.73$, $P < 0.001$]. There was no difference in the number of breed choices reported by participants in the two counterpart control conditions [$M_{lucky\ control\ group}$ = 4.48 vs. $M_{unlucky\ control\ group}$ = 4.80; $F(1, 125) = 1.43$, $P > 0.05$]. There was a significant difference in the number of varietal choices between the lucky group and the corresponding control group [$F(1, 113) = 43.44$, $P < 0.001$]. There was no significant difference in the number of varieties selected between the unlucky group and the corresponding control group [$F(1, 148) = 2.45$, $P > 0.05$] (Table 2).

*Number of brand choices.* The results indicated that the number of brands chosen by the lucky group was greater than the number chosen by the unlucky group [$M_{lucky}$ = 3.56, SD = 0.56 vs. $M_{unlucky}$ = 3.12, SD = 0.67; $F(1, 136) = 16.52$, $P < 0.001$]. There was no difference in the number of brand choices reported by participants in the two counterpart control

**Table 2. Luck and variety-seeking.**

| Indicators | Groups | | | |
|---|---|---|---|---|
| | Lucky | Lucky control | Unlucky | Unlucky control |
| Number of varieties | 6.40 | 4.48 | 4.51 | 4.80 |
| Number of brands | 3.56 | 3.05 | 3.12 | 2.96 |
| Product selection ratio | 0.22 | 0.32 | 0.31 | 0.35 |
| Brands selection ratio | 0.35 | 0.47 | 0.45 | 0.46 |

conditions [M $_{lucky\ control\ group}$ = 3.05 vs. M $_{unlucky\ control\ group}$ = 2.96; F(1, 125) = 0.46, $P$ > 0.05]. There was a significant difference in the number of brand choices between the lucky group and the corresponding control group [F(1, 113) =27.38, $P$ < 0.001]. There was no significant difference in the number of brand choices between the unlucky group and the corresponding control group [F(1, 148) =0.30, $P$ > 0.05].

*Ratio of favorite product purchases to total purchases.* The results showed that the choice ratios of the lucky group were smaller than those of the unlucky group [M $_{lucky}$ = 0.22, SD = 0.08 vs. M $_{unlucky}$ = 0.31, SD = 0.06; F(1, 136) = 45.14, $P$ < 0.001]. There was no difference in participants' reported choice ratios between the two counterpart control conditions [M $_{lucky\ control\ group}$ = 0.32 vs. M $_{unlucky\ control\ group}$ = 0.35; F(1, 125) = 1.15, $P$ > 0.01]. There was a significant difference in the selection ratio between the lucky group and the corresponding control group [F(1, 113) =24.53, $P$ < 0.001]. There was no significant difference in the selection ratio between the unlucky group and the corresponding control group [F(1, 148) = 0.23, $P$ > 0.05].

*Ratio of favorite brand purchases to total purchases.* The results indicate that the choice ratio in the lucky group was smaller than that in the unlucky group [M $_{lucky}$ = 0.35, SD = 0.07 vs. M $_{unlucky}$ = 0.45, SD = 0.06; F(1, 136) = 78.40, $P$ < 0.001]. There was no difference in participants' reported choice ratios between the two counterpart control conditions [M $_{lucky\ control\ group}$ = 0.47 vs. M $_{unlucky\ control\ group}$ = 0.46; F(1, 125) = 0.15, $P$ > 0.05]. There was a significant difference in the selection ratio between the lucky group and the corresponding control group [F(1, 113) =36.35, $P$ < 0.001]. There was no significant difference in the selection ratio between the unlucky group and the corresponding control group [F(1, 148) =2.64, $P$ > 0.05].

## Discussion

The results of experiment 1 supported Hypothesis 1. Lucky numbers made subjects very lucky and led to an increase in variety-seeking. Subjects in the lucky group showed more preference for variety-seeking relative to the control and unlucky groups. The number of varieties and brands chosen by the lucky group was greater than those chosen by the unlucky and control groups, and the ratio of choices of favorite varieties and brands in the lucky group was smaller than the ratio of choices in the control and unlucky groups.

Luck priming can temporarily affect their luck perception and influence consumer variety-seeking, implying a change in subjects' self-perception. Experiment 2 verified the role of novelty-seeking motivation in the effect of luck perception on variety-seeking after luck priming.

## Experiment 2 mediating role of novelty-seeking motivation

Luck gives consumers a psychological force that creates a strong novelty-seeking motivation to prefer variety-seeking. The purpose of this experiment is to test the mediating effects of novelty-seeking motivation in the influence of luck perception on variety-seeking. Additionally, Jiang et al. argued that emotions may also be a mechanism by which luck influences consumer behavior [16]. For example, Darke and Freedman verified that luck experiences can generate positive emotions such as happiness and optimism in consumers [2]. Therefore, this experiment tested the substitution mechanism of happy and optimistic emotions.

### Methods

The research was approved by the Medical Service Center of East China University of Technology (ethics approval number 24019). 20/03/2023~21/03/2023. We recruit participants using the Credamo Research Platform (www.credamo.com), and require them to have independent consumption experience. One hundred and forty-four consumers participated in the experiment, with a mean age of 35.32 years (SD = 4.93); 56 were male, accounting for 38.89%. A single factor (type of priming: lucky vs. unlucky) experiment was used, and participants were randomly assigned to any group. There were 72 each in the lucky and unlucky groups.

Firstly, in order to prime participants' luck perception, we designed a lucky draw, each winner was rewarded with five dollars. Each participant had one draw. The Big Wheel Lucky Draw software was used [20]. Participants in the lucky group were told that the probability of winning the lottery was 0.1 (actually the probability of winning was set to 1, which meant that each subject could win the lottery). The unlucky group was told that the probability of winning the prize was 0.5 (actually the probability of winning was set to 0, which meant that each subject could not win the prize). After participants in the lucky group drew their prizes, they were presented with a 5-second message as follows: "You are lucky. Congratulations! You have won the prize." The unlucky group also received a 5-second message: "Unfortunately, you have not won the prize."

After the lucky draw, the participants were asked to complete the "Shopping Preferences" survey. The survey had two parts. Part one investigated participants' variety-seeking, focusing on their interest in product or brand variety (vs. consistency), and asked whether they would prefer the same or different products in the following consumption scenarios over the next few days (based on Fishbach et al.'s methodology [21]). Specifically, the participants were asked to rate their level of interest in the following behaviors on a 7-point Likert scale with 1 = "same" and 7 = "different" and (2) and (4) reverse scored: (1) 1 large 1L bottle of the same shampoo vs. 5 small 0.2L bottles of a different shampoo; (2) a music CD from a different artist vs. a music CD from the same artist; (3) staying in the same hotel in the same city vs. staying in a different hotel in the same city; (4) visiting a different city during a European visit vs. visiting the same city in depth; (5) shopping in the same shop vs. shopping in a different shop.

Part two investigated participants' novelty-seeking motivation, luck perception, happiness, and optimism. Novelty-seeking motivation was referenced to Choi and Johnson's measure [19], with eight items (α = 0.85) including sentences such as "I am open to receiving information about unfamiliar products or brands" and "I am willing to go to places where I can access information about unfamiliar products or brands"; luck perception measure: "I feel lucky today"; happy mood: "I am feeling happy now"; and optimism: "I am optimistic right now." The experiments used the 7-point Likert scale (1 = "strongly disagree" and 7 = "strongly agree"). After completing this survey, the participants were asked to provide feedback on the survey. The feedback indicated that none of the subjects were able to guess the purpose of the experiment.

## Results

The priming effect of participants' luck perception was examined. The lucky group rated higher than the unlucky group [$M_{lucky}$ = 5.08, SD = 1.67 vs. $M_{unlucky}$ = 3.26, SD = 1.26; $F(1, 142) = 54.33$, $P < 0.001$], indicating that the use of the lucky draw stimulus was very effective. Additionally, the novelty-seeking motivation in the lucky group was stronger than that in the unlucky group [$M_{lucky}$ = 4.95, SD = 0.91 vs. $M_{unlucky}$ = 3.58, SD = 0.98; $F(1, 142) = 75.53$, $P < 0.001$]. The gladness ratings in the lucky group were higher than those in the unlucky group [$M_{lucky}$ = 5.31, SD = 1.51 vs. $M_{unlucky}$ = 4.03, SD = 1.39; $F(1, 142) = 27.89$, $P < 0.001$]. Optimistic ratings were higher in the lucky group than those in the unlucky group [$M_{lucky}$ = 5.26, SD = 1.49 vs. $M_{unlucky}$ = 4.15, SD = 1.85; $F(1, 142) = 15.73$, $P < 0.001$].

Secondly, the effect of priming type on participants' variety-seeking was examined. The analysis of variance showed that the lucky group had a greater preference for variety-seeking relative to the unlucky group [$M_{lucky}$ = 3.83, SD = 0.87 vs. $M_{unlucky}$ = 3.16, SD = 0.67; $F(1, 142) = 27.46$, $P < 0.001$] (Fig 1). Regarding specific choices, the lucky group had a higher variety of shampoo choices than the unlucky group [$M_{lucky}$ = 3.79, SD = 1.23 vs. $M_{unlucky}$ = 3.14, SD = 0.88; $F(1, 142) = 13.41$, $P < 0.01$]. The variety-seeking choices of music in the lucky group were higher than those in the unlucky group [$M_{lucky}$ = 4.01, SD = 1.17 vs. $M_{unlucky}$ = 3.13, SD = 0.93; $F(1, 142) = 25.43$, $P < 0.001$]. There was no significant difference between the lucky group and the unlucky group, although the lucky group had more variety-seeking choices of tourist cities [$M_{lucky}$ = 3.27, SD = 0.94 vs. $M_{unlucky}$ = 3.11, SD = 0.81; $F(1, 142) = 1.30$, $P > 0.05$]. The lucky group's variety-seeking choices of hotels were higher than those of the unlucky group [$M_{lucky}$ = 4.06, SD = 1.12 vs. $M_{unlucky}$ = 3.18, SD = 0.97; $F(1, 142) = 22.03$, $P < 0.001$]. The lucky group diversified their choices of shopping outlets more than the unlucky group [$M_{lucky}$ = 4.01, SD = 1.14 vs. $M_{unlucky}$ = 3.21, SD = 1.01); $F(1, 142) = 20.12$, $P < 0.001$].

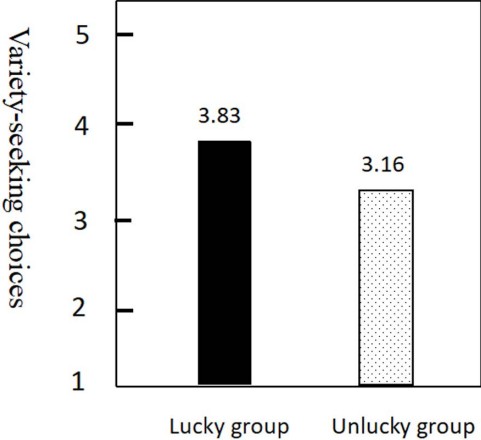

**Fig 1. The Influence of Luck perception on variety-seeking Choices.**

The mediating effect was tested next, using the PROCESS 3.5 plug-in for mediating effects, with a sample size of 5,000 selected, and the percentile bootstrap CI method chosen with a 95% confidence interval. The results showed that the mediating value of novelty-seeking motivation in the effect of priming type on variety-seeking was 0.48 (SD = 0.17, $P < 0.05$, 95% CI: 0.29~0.69). The coefficient of the effect of priming type on novelty-seeking motivation was 1.31 (SD = 0.15, $P < 0.05$). The influence coefficient of novelty-seeking motivation on variety-seeking choice was 0.35 (SD = 0.14, $P < 0.05$) (Fig 2). Novelty-seeking motivation mediated the influence of luck on variety-seeking, which supported Hypothesis 2.

Finally, the substitution mechanism causing gladness and optimism to influence the effect of luck perception on variety-seeking choices was tested. For gladness, the mediating effect value in the effect of priming type on variety-seeking was −0.04 ($P > 0.05$, 95% CI: −0.20 to 0.06), and the mediating effect was not significant. For optimism, the mediation effect value in the effect of priming type on variety-seeking was 0.01 ($P > 0.05$, 95% CI: −0.09~0.10), and the mediation effect was not significant.

## Discussion

Experiment 2 verified that the lucky draw produced higher luck evaluations and evoked the participants' variety-seeking. Among the variety-seeking choices of five products, the lucky group rated the variety-seeking choices of four products significantly higher than the unlucky group, but for travelling cities, although the variety-seeking preference of the lucky group was stronger than that of the unlucky group, it was not significant. The possible reason is that most of the subjects in the experiment are Asian, most of them never travelled Europe and had no concept of European cities, leading to a bias

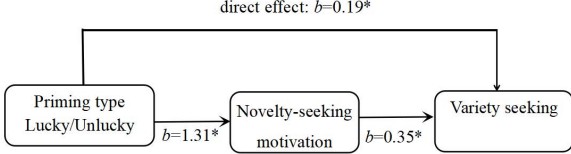

**Fig 2. The mediating effects of novelty-seeking motivation.** Note: *p< < 0.05.

in variety-seeking choices. Novelty-seeking motivation mediated the effect of luck perception on variety-seeking behavior, but optimism and a happy mood did not mediate, suggesting that mood does not explain the effect of luck perception on variety-seeking. Additionally, luck is primed by words, lucky numbers, and an initial lucky draw. Thus, experiment 3 used an idiomatic word-filling game to prime participants' luck perception.

## Experiment 3 moderating role of need for cognitive closure

Tao et al. argues that there are boundary conditions for the effects of luck on consumer behavior, and there are differences in the luck effects produced in different types of individuals [20]. This experiment tested the moderating effect of need for cognitive closure in the effect of luck perception on variety-seeking.

### Methods

The research was approved by the Medical Service Center of East China University of Technology (ethics approval number 24019). A 2 (type of priming: lucky vs. unlucky) × 2 (need for cognitive closure: high vs. low) between-groups experimental design was used. 24/07/2023~26/07/2023. The subjects are required to have independent consumption experience. A total of 224 customers participated in the experiment, with a mean age of 36.99 years (SD = 6.92); 67 or 29.91% were male. They were randomly assigned to the lucky group (114) and the unlucky group (110) and asked to participate in two ostensibly unrelated experiments. The first was a questionnaire on traits needed for cognitive closure, and the latter was a consumer choice experiment.

Roets and Van Hiel's need for cognitive closure measurement scale with 15 items (a = 0.83) was used [41]. It includes sentences such as, "I feel relieved when I make a decision"; "I dislike questions that can be answered in more than one way"; and "When I encounter a problem, I desire to find a solution quickly."

After the participants completed the need for cognitive closure scale, we used subliminal priming for luck perception. Referring to word priming method [20], the participants were informed that this was a study on an idiomatic crossword puzzle, where each idiom was missing a word, and they were asked to fill in the correct word. After filling in the word, they were asked if they knew the meaning of the idiom, with 0 = "no knowledge" and 1 = "knowledge." With this in mind, the lucky group needed to complete 10 idioms, eight implied good luck (e.g., "When the time comes, it's good luck") and two were neutral idioms (e.g., "Silent" and "silent"). The unlucky group also had to complete 10 idioms, eight implied unlucky (e.g., "bad luck") and two were neutral idioms. Twenty subjects pre-tested these idioms, −3 (very unfortunate) to 3 (very fortunate). There was a difference in luck ratings between the lucky, neutral, and unlucky idioms [$M_{lucky}$ = 1.70 vs. $M_{Neutral}$ = −0.15 vs. $M_{unlucky}$ = −1.15; $F(2, 57)$ = 48.35, $p < 0.001$].

After the priming task was completed, the participants were asked to complete a snack choice task based on Fan and Jiang's method [22]. There were six types of snacks available (e.g., multi-flavoured peanuts, crisps, thin crackers, melon seeds, chocolate, and sultanas). The participants were asked to choose a snack for the next five weeks, and they could choose only one snack per week. Each snack could be repeated, so the number of types of snacks the participants choose (up to 5) can be regarded as their tendency to engage in variety-seeking.

### Results

The idioms chosen were frequently used in daily life, so the participants were able to answer correctly. Additionally, the participants were able to understand the meaning of each idiom.

The participants' luck perception was examined, and the lucky group had higher luck perception ratings than the unlucky group [$M_{lucky}$ = 5.18, SD = 1.23 vs. $M_{unlucky}$ = 3.53, SD = 2.02; $F(1, 222)$ = 54.92, $P < 0.001$], which suggests that the method of the Idiom Fill-In Game can successfully manipulate their luck perception. Furthermore, the lucky group had higher happy ratings than the unlucky group [$M_{lucky}$ = 5.32, SD = 1.29 vs. $M_{unlucky}$ = 4.58, SD = 1.69; $F(1, 222)$ = 13.41, $P < 0.001$]. The optimistic ratings of the lucky group were higher than those of the unlucky group [$M_{lucky}$ = 5.21, SD = 1.43

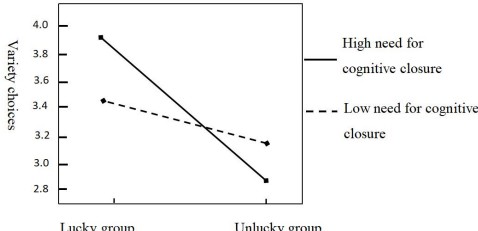

**Fig 3. The influence of luck on variety-seeking under different need for cognitive closure.**

vs. $M_{unlucky}$ = 4.69, SD = 1.75; F (1, 222) =6.16, $P$ < 0.05]. Variance analyses showed that the lucky group had a higher choice of snack variety than the unlucky group [$M_{lucky}$ = 3.74, SD = 1.11 vs. $M_{unlucky}$ = 3.08, SD = 1.23; F (1, 222) = 17.79, $P$ < 0.001].

The interaction of priming type and need for cognitive closure on snack variety choices was analyzed. The moderating effect of the need for cognitive closure was examined using the PROCESS 3.5 plug-in by selecting model 1, ticking "-1SD, mean, +1SD," with a confidence interval of 95%. The results showed that there was a significant effect of priming type (1 for lucky, 0 for unlucky) on snack variety choice with a value of 0.66 ($p$ < 0.05, 95% CI: 0.35~0.97). The need for cognitive closure had no significant effect on snack variety choice with an effect value of 0.05 ($p$ > 0.05, 95% CI: −0.13 to 0.23).

The interaction effect between priming type and need for cognitive closure on snack variety choice was significant with an effect value of 0.40 ($p$ < 0.05, 95% CI: 0.04~0.76), as shown in Fig 3. For participants with high need for cognitive closure, there was a significant effect of priming type on snack variety choice with an effect value of 1.01 ($p$ < 0.05, 95% CI: 0.57~1.44). That is, for participants with high need for cognitive closure, priming lucky perception will produce variety-seeking choices. For participants with low need for cognitive closure, the effect value of priming type on snack variety choice was 0.31 ($p$ > 0.05, 95% CI: −1.12~0.75), and there was no significant effect. That is, there was no significant difference in variety-seeking choice for participants with low need for cognitive closure, whether good luck was primed or not.

## Discussion

Experiment 3 supported Hypothesis 1 and Hypothesis 3. Luck perception has a positive effect on consumer variety-seeking, and need for cognitive closure moderate that effect. For consumers with a low need for cognitive closure, lucky or unlucky perception does not have a significant effect on their variety-seeking preferences, whereas for consumers with a high need for cognitive closure, having lucky perception gives them a stronger preference for variety-seeking.

## Results and discussion

Luck is an integral experience in our daily lives. Luck affects our emotions and cognition [16] and influences behavioral decisions, especially in competitive sports, lotteries and economic decisions, but researchers have paid little attention to the study of the impact of luck on consumer behavior [42]. Based on the psychological power theory of luck, this paper demonstrates the effect of luck perception on variety-seeking from a risk perspective, enriching the research on luck in consumer behavior.

## Conclusion

This paper found that luck perception positively affects consumer variety-seeking. When making consumption choices, consumers who feel lucky have more variety-seeking tendencies, choose more brands and products, and prefer to

consume different goods at different times and places. More research found that novelty-seeking motivation is the mechanism by which luck perception influences variety-seeking. To prime luck perception, consumers generate strong novelty-seeking motivation, prefer novel and unfamiliar consumption choices, and are more willing to choose variety-seeking. Additionally, luck cognition is a kind of latent consciousness preexisting in the brain, which is easily activated by luck cues. Thus, the influence of luck perception is moderated by individual cognitive traits. For customers with high need for cognitive closure, having luck perception produced more variety-seeking, but for customers with low need for cognitive closure, having or not having luck perception had no significant effect on their variety-seeking preferences.

## Theoretical contribution

Firstly, this paper reaffirms the validity of luck priming. The current common subliminal priming procedures for luck are number priming and word priming [16,20]. Based on the previous work, this paper combined the Chinese luck culture, cleverly designed the luck number and luck idiom stimulus materials, and confirmed that the method could efficiently activate the participants' sense of luck. In Experiments 1 and 2, the experimental group's luck was significantly higher than those of the control and unlucky groups. Additionally, the lucky draw game experience in Experiment 2 primed the participants' luck, thus, the game can provide effective and reliable activation materials for future experimental research on luck.

Secondly, this paper enriches the research on luck in risky choice behavior. Previous studies have shown that participants with luck have a positive preference for behavioral decisions even when other consumers are uncertain about information. For example, consumers with luck have a positive affective bias towards risk and have increased risky behaviors [5], such as purchasing lottery tickets more frequently and in varied forms [16,30,31]. This paper investigates the influence of luck perception on consumers' decision-making, where luck leads them to more variety-seeking. In fact, this paper continues previous research on the association of luck with risk. Consumer decision-making involves uncertainty and ambiguity, and uncertainty in consumer decision-making can be subdivided into information uncertainty and information incompleteness [23]. Information uncertainty refers to the probability of the occurrence of an event involved in the information, and information incompleteness refers to the failure to obtain comprehensive information about consumer decision-making. Variety-seeking can stimulate consumers' excitement and novelty [8]. Variety-seeking occurs when consumers try or seek out a new brand or product without being familiar with information about the product or brand. This behavior suffers from a large degree of missing information and can be regarded as purchasing behavior in the presence of incomplete information. However, purchasing decisions based on information uncertainty and incompleteness have similar influence mechanisms [23]. Therefore, this paper confirms that luck has a similar influence on purchase decisions with uncertain and incomplete information, extends the influence of luck to variety-seeking with incomplete information, and enriches the study of luck in risky choice behaviors.

Thirdly, this paper provides an in-depth analysis of self-concept change, which is the underlying mechanism influencing the impact of luck on consumer behavior. According to the positive self-concept explanation, luck perception may affect active self-perception, and this transient change in self-concept is the reason luck has an impact on individual behavior. For example, luck can influence individual behavior through changes in self-efficacy, sense of control, etc. This paper discovered that luck perception affects novelty-seeking motivation, changes in this novelty-seeking motivation affect consumer variety-seeking, and novelty-seeking motivation is the mechanism by which the sense of luck perception affects variety-seeking. Additionally, this paper found that luck positively affects consumers' emotions, but these emotional changes are not the reason luck affects variety-seeking. This does not mean that positive emotions do not work, but the influence of emotions is more prevalent. This also validates Darke and Freedman's [2] hypothesis that luck causes individual to become more optimistic and does not affect their uncertainty behaviors but makes them rely more on familiar behaviors. Another possible explanation is that there may be an inverted U-shaped relationship between the effects of positive emotions on variety-seeking [8]. When positive emotions are strong, individuals' variety-seeking decreases, leading to a non-significant effect of positive emotions on variety-seeking. By excluding the effect mechanism of emotion and

confirming the effect mechanism of novelty-seeking motivation, we have provided an in-depth theoretical and empirical exploration of how the luck perception affects consumer variety-seeking and enriched the research on the mechanism of luck's influence on consumer behavior.

Finally, this paper introduces individual traits into the research framework and identifies boundary conditions under which individual need for cognitive closure work effectively. It has been shown that the effect of luck on consumption behavior is not always in the same direction. For example, Tao et al. found an interaction between trait-based luck beliefs and state-based luck perception [20]. For consumers with weak luck beliefs, luck priming has no significant impact on their decision-making behavior. This paper similarly discovered that another trait of consumers, namely need for cognitive closure, also moderates the effect of luck perception on consumption behavior. The effect of luck on variety-seeking is only significant among those with high need for cognitive closure. This establishes boundary conditions for the study of the effect of luck on variety-seeking, and constructs a more in-depth and clearer framework in both theoretical and applied areas.

## Practical implications

Luck is increasingly being used by businesses as an actionable marketing element to influence consumer behavior. These firms want to influence consumers' consumption decisions by improving their luck perception. Therefore, This paper's findings have managerial implications for business managers or marketers.

Firstly, the findings have implications for firms expanding their product lines and launching new brands. When enterprises launch new products or brands, enterprise managers or marketers can consider altering consumers' luck perception to influence their consumption decisions. For instance, they can incorporate lucky numbers or symbols on product packaging or during advertising campaigns. When promoting products or brands offline, they can utilize lucky projection techniques or lucky draws to stimulate consumers' luck perception, fostering a greater variety of demands and enhancing their preference for new products or brands, thereby achieving market promotion for new products and brands.

Secondly, the research conclusions assist enterprises in implementing precise marketing. Enterprise managers or marketers can utilize big data technology, such as by simplifying the need for cognitive closure scale, to simply inquire about consumers' need for cognitive closure and identify their personality traits, thereby tracking and identifying consumers with high need for cognitive closure. When facing these consumers, enterprise managers or marketers can push luck-related information to them while they browse products, stimulating their sense of luck, increasing their variety-seeking behavior, and encouraging them to purchase more different products, thereby increasing the sales volume of the enterprise.

## Limitations

The findings of this paper are limited to the conclusions drawn from the experiments we conducted, and there are some limitations that provide many opportunities for future research.

This paper only tests the explanatory mechanism of novelty-seeking motivation in the effect of luck perception on variety-seeking. Other explanatory mechanisms could be considered in the future. In risky decision making, it has been demonstrated that control illusions and facilitation orientation can explain the role of luck in influencing risky behavior [43], but all these studies have validated luck's influence from a cognitive perspective, whereas emotions have not been explored in depth, except the finding that emotions such as gladness and optimism do not explain the role of luck's influence on some risky behaviors. This is only in the domain we considered, where the influence of luck-induced positive emotions on behavior was not evident. However, emotions may have significant effects on other behavioral domains. For example, positive affect increases individual's social and benevolent behavior [7]. Therefore, future research could verify whether luck perception and positive emotions have an impact on consumers' helping behaviors.

Moreover, luck is a cultural phenomenon. In terms of luck priming effects, participants who chose a strong culture of luck had more pronounced effects than those who chose a less strong culture of luck [16,19,20]. Therefore, in addition to

the influence of high need for cognitive closure traits on the luck priming effect, cultural differences— luck priming's suitability to consumers with Eastern cultural backgrounds or consumers in Western countries—is an equally important determinant of proneness to luck. Future research could examine how cultural differences influence luck and luck's effects. and examine whether luck has an impact on cultural consumption.

## Supporting information

**S1 Data. Compressed/ZIP File Archive.**
(ZIP)

## Author contributions

**Conceptualization:** Jianbin Zhao.

**Investigation:** Jianbin Zhao.

**Methodology:** Jianbin Zhao.

**Writing – original draft:** Zheng Li.

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
