## [Decision Letter · Decision Letter 0]

Dear Dr. Zhao,

We look forward to receiving your revised manuscript.

Kind regards,

Alessandro Pluchino

Academic Editor

PLOS ONE

Journal Requirements:

“This study was supported by the National Natural Science Foundation of China (71962001); Humanities and Social Sciences Research Foundation of Ministry of Education (21YJC630069); Guangdong superior agricultural products foreign trade innovation team (2020WCXTD013); Greater Bay Area Agricultural products Circulation Research Center Study on ways and countermeasures to enhance competitiveness of aquatic products in Guangdong Province (2022ZDJS018).”

5. We note that your Data Availability Statement is currently as follows: “All relevant data are within the manuscript and in Supporting Information files.”

6. Please amend either the title on the online submission form (via Edit Submission) or the title in the manuscript so that they are identical.

Reviewers' comments:

Reviewer's Responses to Questions

**Comments to the Author**

1. Is the manuscript technically sound, and do the data support the conclusions?

Reviewer #1: Yes

Reviewer #2: Yes

2. Has the statistical analysis been performed appropriately and rigorously?

Reviewer #1: Yes

Reviewer #2: Yes

3. Have the authors made all data underlying the findings in their manuscript fully available?

Reviewer #1: Yes

Reviewer #2: Yes

4. Is the manuscript presented in an intelligible fashion and written in standard English?

Reviewer #1: Yes

Reviewer #2: Yes

Reviewer #1: Reviewer Attachments for Manuscript Number PONE-D-24-49678

The effects of luck perception on consumer variety-seeking: The mediating role of novelty-seeking motivation and the moderating role of need for cognitive closure

General Assessment

1. Originality and Contribution to the Field

The manuscript explores the under-researched area of luck perception in consumer behavior, offering valuable insights into its psychological underpinnings and practical implications.

Recommendation: Highlight the novelty of examining luck perception in the context of variety-seeking and its practical relevance to marketing strategies. Discuss how these findings fill gaps in existing research on consumer behavior and psychological motivations.

2. Clarity and Structure

The manuscript is generally well-organized, but the dense presentation of experimental methods and statistical analyses may be challenging for readers unfamiliar with these techniques.

Recommendation: Simplify the descriptions of experimental designs and statistical methods. Consider adding summaries or diagrams to make complex concepts more accessible.

3. Theoretical Framework

The manuscript presents a solid theoretical foundation, but the discussion of novelty-seeking motivation and the need for cognitive closure could be more comprehensive.

Recommendation: Provide a deeper discussion of novelty-seeking and cognitive closure theories. Explain their relevance to consumer behavior and their integration with luck perception.

Key Concerns and Recommendations

1. Methodology

The experimental designs are robust, but some aspects of the methodology, such as participant selection, priming techniques, and statistical assumptions, require further elaboration.

Recommendation: Clearly explain the criteria for participant recruitment and how cultural factors might influence responses to luck perception. Justify the choice of priming methods and discuss potential biases.

2. Operationalization of Key Variables

Key variables like luck perception, novelty-seeking motivation, and need for cognitive closure are defined but require more detail on their measurement and validation.

Recommendation: Provide detailed descriptions of scales used to measure these variables, including their reliability and validity. Explain how the experimental manipulations were verified.

3. Results Interpretation

The results are well-presented, but their practical implications are underexplored.

Recommendation: Discuss the real-world applications of these findings, such as how marketers can use luck perception to design campaigns that encourage variety-seeking. Provide specific examples of potential strategies.

Moderation and Mediation Effects

While the moderating and mediating effects are statistically significant, their broader implications are not fully discussed.

Recommendation: Elaborate on how the need for cognitive closure influences the relationship between luck perception and variety-seeking. Discuss the implications of novelty-seeking motivation as a mediator in consumer decision-making.

Overall Recommendation Major Revision Required

To ensure the manuscript meets the standards of PLOS ONE, it requires substantial revisions. These revisions should focus on clarifying the methodology, deepening the interpretation of results, and expanding on both theoretical and practical implications. If these revisions are thoroughly addressed, the manuscript has the potential to make a significant contribution to the study of followership and leadership communication.

Detailed Comments on Manuscript Sections

1. Title and Abstract

Title

Comment: The title is clear and informative but could emphasize the experimental nature of the study.

Recommendation: Consider revising the title to reflect the experimental approach (e.g., "Experimental Evidence on the Role of Luck Perception in Consumer Variety-Seeking").

Comment: The abstract effectively summarizes the study but lacks emphasis on practical implications.

Recommendation: Include a brief discussion of how these findings could inform marketing practices or consumer engagement strategies.

2. Introduction

Comment : The introduction provides a strong rationale for the study but could better contextualize the research within broader consumer behaviour literature.

Recommendation: Add a discussion of how this research complements or extends existing studies on psychological influences in consumer decision-making.

3. Literature Review

Comment: The review covers key concepts but does not fully integrate them into a cohesive framework.

Recommendation: Expand on the relationships between luck perception, novelty-seeking motivation, and variety-seeking. Include additional references to recent studies in consumer psychology.

4. Methodology

Comment: The explanation of structural equation modeling (SEM) is dense and may be difficult for readers unfamiliar with advanced statistical techniques. While the fit indices are reported, there is little explanation of what these indices mean or why they indicate an acceptable model fit. Additionally, the assumptions underlying SEM (such as measurement invariance and suppression effects) are mentioned but not clearly explained.

Recommendation: The methodology is detailed but dense, making it challenging for readers to follow.

5. Results

Comment: The results are statistically robust but lack sufficient interpretation.

Recommendation: Highlight the practical significance of findings, such as how marketers can leverage these insights to design interventions that enhance variety-seeking.

6. Discussion

Comment: The discussion effectively summarizes the results but does not fully explore their theoretical and practical implications.

Recommendation: Provide a deeper analysis of how these findings contribute to understanding consumer psychology. Discuss potential applications in marketing and product design.

7. Conclusion and Future Directions

Comment: The conclusion is concise but could offer more specific suggestions for future research.

Recommendation: Suggest investigating cultural differences in luck perception or exploring its impact on other consumer behaviors. Discuss how longitudinal studies could extend these findings.

8. Final Comments (Minor Revision)

The manuscript addresses an important topic in consumer psychology and marketing. However, to enhance its clarity and impact, revisions are needed to elaborate on methodological details, deepen theoretical discussions, and explore practical implications. With these improvements, the study has the potential to make a significant contribution to the field.

Reviewer #2: The paper is technically sound and highly relevant, exploring an interesting and practical topic—the effect of luck perception on consumer variety-seeking. The authors clearly structured their research, carefully crafted their hypotheses based on existing literature, and employed rigorous experimental methods across three distinct studies. Their statistical analyses robustly support the conclusions, effectively demonstrating that luck perception increases consumers' variety-seeking behaviors through novelty-seeking motivation, especially among consumers with a high need for cognitive closure. Importantly, the authors have transparently made their underlying data fully available within the manuscript, enhancing the credibility of their findings. Although minor improvements in language and clarity could enhance readability, the paper offers valuable theoretical contributions and practical insights, particularly for marketers aiming to strategically leverage perceptions of luck. Overall, the manuscript is scientifically sound, methodologically rigorous, and clearly conveys meaningful insights relevant to both researchers and marketing practitioners.

**Do you want your identity to be public for this peer review?** For information about this choice, including consent withdrawal, please see our Privacy Policy

Reviewer #1: No

Reviewer #2: No

---

## [Author Response · Author response to Decision Letter 1]

17 Apr 2025

Modification instructions

Dear Reviewers:

I would like to thank you sincerely for your valuable comments on this paper, which were very professional and pertinent, pointing out shortcomings and giving suggestions for revision, which have been very enlightening. Your suggestions have been very helpful and we. The authors have followed your comments to read and revise the whole text again and again and to explain it in detail.

Reviewer #1

General Assessment

1. Originality and Contribution to the Field. The manuscript explores the under-researched area of luck perception in consumer behavior, offering valuable insights into its psychological underpinnings and practical implications. Recommendation: Highlight the novelty of examining luck perception in the context of variety-seeking and its practical relevance to marketing strategies. Discuss how these findings fill gaps in existing research on consumer behavior and psychological motivations.

Revision note: Thank you very much for your suggestions.

We have summarized the research significance of this paper as follows:

“In this paper, the influence of luck on variety-seeking was explored in depth through experimental research methods. The main research questions include the direct influence of luck on variety-seeking, the mechanism of novelty-seeking motivation in the effect of luck on variety-seeking, and the boundary conditions need for cognitive closure. The findings offer innovative insights into the influence of luck in the field of consumption, broadening the scope of research on luck effects in product or brand decision-making. Furthermore, they delve into the psychological effects of luck, providing valuable theoretical support for people to gain a scientific understanding of luck. From a practical viewpoint, this paper’s findings will shed light on the expansion of product lines and the launch of new brands and provide guidance for enterprises to implement precision marketing as well as luck marketing.”

2. Clarity and Structure. The manuscript is generally well-organized, but the dense presentation of experimental methods and statistical analyses may be challenging for readers unfamiliar with these techniques. Recommendation: Simplify the descriptions of experimental designs and statistical methods. Consider adding summaries or diagrams to make complex concepts more accessible.

Revision note: Thank you very much for your suggestions. This paper is written in accordance with the norms for psychological experiments, employing standardized terminology in the experimental section. If the processes and designs are overly simplified, readers may struggle to comprehend the entire experimental procedure, rendering it unhelpful as a reference for similar experimental designs in the future.

3. Theoretical Framework. The manuscript presents a solid theoretical foundation, but the discussion of novelty-seeking motivation and the need for cognitive closure could be more comprehensive. Recommendation: Provide a deeper discussion of novelty-seeking and cognitive closure theories. Explain their relevance to consumer behavior and their integration with luck perception.

Revision note: Thank you very much for your questions and suggestions.

We have reorganized the logical reasoning of the hypothesis to make the explanation of novelty-seeking motivation and the need for cognitive closure more comprehensive. The specifics are as follows:

“Research on luck suggests that consumers who feel lucky become more energetic and possess more preferences for unfamiliar or novel choices. For example, consumers who feel lucky are more likely to use online banking, rather than a physical bank, to deposit money (Jiang et al., 2009). In financial markets, consumers who feel lucky have a higher interest in sudden and unfamiliar market information; they are more willing to pay for random shock signals in the lottery market (Gao et al., 2021). It is evident that luck motivates consumers and develops their interest in and desire for novel and unknown things. Novelty seeking is the innate tendency of consumers to show pleasure or excitement for novel stimuli (Thomas-Walters, 2021).

Based on the above analysis, we argue that consumers who feel lucky possess a strong motivation to pursue novelty.

The motivation for novelty seeking is one of the important factors influencing individual behavior. It represents consumers' innate tendency to exhibit pleasure or excitement towards novel stimuli, and serves as one of the driving forces for consumers to actively pursue new experiences (Nguyen et al., 2020). This is manifested in two aspects: intrinsic novelty seeking is the individual's desire for novel and exciting stimuli; extrinsic novelty seeking is manifested in the individual's actual behavior to obtain new stimuli. Therefore, we believe that the mechanism of luck’s influence on variety-seeking emerges from consumer novelty-seeking motivation. This is because variety-seeking is an expression of consumers’ search for novelty and pursuit of maximizing psychological utility, which can satisfy consumers’ novelty needs (Cao & Wang 2018). If consumers repeatedly choose a product, their stimulation for the product becomes weaker and fails to satisfy their novelty needs. on the other hand, found that when consumers have strong novelty-seeking motivation, they will change their routine purchasing behaviors, such as choosing new and unfamiliar products or brands. Meanwhile, when their novelty-seeking motivation is weaker, they will continue to maintain routine buying behavior (Sahni & Gupta, 2019).”

“Cognitive closure needs, as a motivational factor influencing individuals' ambiguous decision-making in uncertain situations, are composed of dimensions such as cognitive structuring, ambiguity aversion, decision decisiveness, expectation predictability, and psychological closure. Cognitive closure needs are both personality traits and context-dependent, and can be aroused by time pressure, psychological fatigue, environmental noise, and task attractiveness. Individuals with higher cognitive closure needs often follow the principles of decision immediacy and decision continuity, and have a strong tendency towards grasping and freezing. Cognitive closure needs affect individuals' attitudes towards ambiguity and information processing methods. Those with higher cognitive closure needs have lower ambiguity tolerance and are more prone to heuristic information processing; whereas consumers with lower cognitive closure needs have higher ambiguity tolerance and are more likely to engage in analytical information processing. Individuals with high cognitive closure needs rely more on conventions or stereotypes during information processing, less on information search, and are prone to forming larger decision biases; whereas those with low cognitive closure needs have higher ambiguity tolerance during information processing, rely more on information search, and therefore make more accurate decisions. These research conclusions have been widely tested in the fields of social judgment, team interaction, consumer behavior, negotiation, and politics.”

4. Methodology. The experimental designs are robust, but some aspects of the methodology, such as participant selection, priming techniques, and statistical assumptions, require further elaboration. Recommendation: Clearly explain the criteria for participant recruitment and how cultural factors might influence responses to luck perception. Justify the choice of priming methods and discuss potential biases.

Revision note: Thank you very much for your questions and suggestions. We have made further improvements to address the deficiencies in this area. The specifics are as follows:

Experiment 1�“The subjects are required to have independent consumption experience. The experiment used a single factor analysis method (type of priming: lucky vs. unlucky vs. control), and participants were randomly assigned to the lucky (63), unlucky (75), and control (127) groups.”

Experiment 2�“We recruit participants using the credamo research platform (www.credamo.com), and require them to have independent consumption experience. one hundred and forty-four consumers participated in the experiment, with a mean age of 35.32 years (SD = 4.93); 56 were male, accounting for 38.89%. A single factor (type of priming: lucky vs. unlucky) experiment was used, and participants were randomly assigned to any group. There were 72 each in the lucky and unlucky groups.”

Experiment 3�“The subjects are required to have independent consumption experience. a total of 224 customers participated in the experiment, with a mean age of 36.99 years (SD = 6.92); 67 or 29.91% were male. They were randomly assigned to the lucky group (114) and the unlucky group (110) and asked to participate in two ostensibly unrelated experiments. The first was a questionnaire on traits needed for cognitive closure, and the latter was a consumer choice experiment.”

In addition, the methods for initiating luck perception all refer to mature methods Jiang et al., 2009; Li, 2021; Zhao & Zhong, 2022), which have been applied in the literature. Therefore, I believe there is no need to further discuss the feasibility of these methods.

Reference

Jiang, Y., Cho, A., & Adaval, R. (2009). The unique consequences of feeling lucky: Implications for consumer behavior. Journal of Consumer Psychology, 19(2), 171-184.

Li, H. (2021). Confidence charms: How superstition influences overconfidence bias in Han and the Qiang ethnic minority Chinese. The Journal of Psychology, 155(5), 473-488.

Zhao, J., & Zhong, H. (2022). The licensing effect of luck: The influence of perceived luck on green consumption intention. Current Psychology, 42(15), 1-15.

5. Operationalization of Key Variables. Key variables like luck perception, novelty-seeking motivation, and need for cognitive closure are defined but require more detail on their measurement and validation. Recommendation: Provide detailed descriptions of scales used to measure these variables, including their reliability and validity. Explain how the experimental manipulations were verified.

Revision note: Thank you very much for your questions and suggestions. According to existing research, researchers generally only report reliability, while reporting on validity is relatively rare. In this experiment, there is only one scale, so it is impossible to conduct a validity analysis.

In addition, this experiment utilized mature priming methods, and the results have proven these methods to be feasible. We tested luck perception in three experiments and found that the experimental group had a higher perception of luck compared to the control group, indicating that the experimental priming was effective.

6. Results Interpretation. The results are well-presented, but their practical implications are underexplored. Recommendation: Discuss the real-world applications of these findings, such as how marketers can use luck perception to design campaigns that encourage variety-seeking. Provide specific examples of potential strategies.

“Luck is increasingly being used by businesses as an actionable marketing element to influence consumer behavior. These firms want to influence consumers’ consumption decisions by improving their luck perception. Therefore, This paper’s findings have managerial implications for business managers or marketers.

Firstly, the findings have implications for firms expanding their product lines and launching new brands. When enterprises launch new products or brands, enterprise managers or marketers can consider altering consumers' luck perception to influence their consumption decisions. For instance, they can incorporate lucky numbers or symbols on product packaging or during advertising campaigns. When promoting products or brands offline, they can utilize lucky projection techniques or lucky draws to stimulate consumers' luck perception, fostering a greater variety of demands and enhancing their preference for new products or brands, thereby achieving market promotion for new products and brands.

Secondly, the research conclusions assist enterprises in implementing precise marketing. Enterprise managers or marketers can utilize big data technology, such as by simplifying the cognitive closure needs scale, to simply inquire about consumers' cognitive closure needs and identify their personality traits, thereby tracking and identifying consumers with high cognitive closure needs. When facing these consumers, enterprise managers or marketers can push luck-related information to them while they browse products, stimulating their sense of luck, increasing their variety-seeking behavior, and encouraging them to purchase more different products, thereby increasing the sales volume of the enterprise.”

7. Moderation and Mediation Effects. While the moderating and mediating effects are statistically significant, their broader implications are not fully discussed. Recommendation: Elaborate on how the need for cognitive closure influences the relationship between luck perception and variety-seeking. Discuss the implications of novelty-seeking motivation as a mediator in consumer decision-making.

We discussed in detail how the need for cognitive closure affects the relationship between luck perception and variety-seeking, as well as the mediating role of novelty-seeking motivation in consumer decision-making.

The mediating role of novelty-seeking motivation

Research on luck suggests that consumers who feel lucky become more energetic and possess more preferences for unfamiliar or novel choices. For example, consumers who feel lucky are more likely to use online banking, rather than a physical bank, to deposit money (Jiang et al., 2009). In financial markets, consumers who feel lucky have a higher interest in sudden and unfamiliar market information; they are more willing to pay for random shock signals in the lottery market (Gao et al., 2021). It is evident that luck motivates consumers and develops their interest in and desire for novel and unknown things. Novelty seeking is the innate tendency of consumers to show pleasure or excitement for novel stimuli (Thomas-Walters, 2021).

Based on the above analysis, we argue that consumers who feel lucky possess a strong motivation to pursue novelty.

The motivation for novelty seeking is one of the important factors influencing individual behavior. It represents consumers' innate tendency to exhibit pleasure or excitement towards novel stimuli, and serves as one of the driving forces for consumers to actively pursue new experiences (Nguyen et al., 2020). This is manifested in two aspects: intrinsic novelty seeking is the individual's desire for novel and exciting stimuli; extrinsic novelty seeking is manifested in the individual's actual behavior to obtain new stimuli. Therefore, we believe that the mechanism of luck’s influence on variety-seeking emerges from consumer novelty-seeking motivation. This is because variety-seeking is an expression of consumers’ search for novelty and pursuit of maximizing psychological utility, which can satisfy consumers’ novelty needs (Cao & Wang 2018). If consumers repeatedly choose a product, their stimulation for the product becomes weaker and fails to satisfy their novelty needs. on the other hand, found that when consumers have strong novelty-seeking motivation, they will change their routine purchasing behaviors, such as choosing new and unfamiliar products or brands. Meanwhile, when their novelty-seeking motivation is weaker, they will continue to maintain routine buying behavior (Sahni & Gupta, 2019).

In sum, luck perception can give consumers a power resource that enhances their novelty-seeking motivation, and variety-seeking can satisfy their novelty-seeking needs. Therefore, luck affects consumer variety-seeking, and novelty-seeking motivation is the mediating mechanism.

Hypothesis 2: Novelty-seeking motivation mediates the influence of luck perception on consumer variety-seeking.

Moderating Role of Need for Cognitive Closure

Research has shown that relative to consumers with low need for cognitive closure, consumers with high need for cognitive closure are cognitively lazy, tend to adopt heuristic, simplified and top-dow

---

## [Decision Letter · Decision Letter 1]

The effects of luck perception on consumer variety-seeking

PONE-D-24-49678R1

Dear Dr. Zhao,

We’re pleased to inform you that your manuscript has been judged scientifically suitable for publication and will be formally accepted for publication once it meets all outstanding technical requirements.

Kind regards,

Alessandro Pluchino

Academic Editor

PLOS ONE

Reviewer's Responses to Questions

**Comments to the Author**

Reviewer #1: All comments have been addressed

2. Is the manuscript technically sound, and do the data support the conclusions?

Reviewer #1: Yes

3. Has the statistical analysis been performed appropriately and rigorously?

Reviewer #1: Yes

4. Have the authors made all data underlying the findings in their manuscript fully available?

Reviewer #1: Yes

5. Is the manuscript presented in an intelligible fashion and written in standard English?

Reviewer #1: Yes

Reviewer #1: I would like to express my sincere appreciation for the authors’ diligent efforts in revising the manuscript and thoroughly addressing each reviewer’s comment. The revised version demonstrates a significant improvement in both content and presentation. Notably, the authors have strengthened the theoretical framework by expanding the discussion on novelty-seeking motivation and the need for cognitive closure, thereby providing a more comprehensive understanding of the psychological mechanisms underlying consumer variety-seeking. The methodological clarifications including participant selection criteria, the justification of priming techniques, and elaboration on statistical procedures have greatly enhanced the transparency and reproducibility of the study. Moreover, the authors’ integration of practical implications, especially in the context of marketing strategies involving luck perception, reflects a meaningful contribution to both academic literature and industry practice. The responses to reviewers were detailed, respectful, and well-articulated, evidencing a high level of scholarly engagement. Overall, this revised manuscript is a testament to the authors’ commitment to research quality and their responsiveness to constructive feedback. I commend the team for their impressive improvements and scholarly rigour.

**Do you want your identity to be public for this peer review?** For information about this choice, including consent withdrawal, please see our Privacy Policy

Reviewer #1: No

---

## [Editor Report · Acceptance letter]

PONE-D-24-49678R1

PLOS ONE

Dear Dr. Zhao,

I'm pleased to inform you that your manuscript has been deemed suitable for publication in PLOS ONE. Congratulations! Your manuscript is now being handed over to our production team.

Kind regards,

on behalf of

Dr. Alessandro Pluchino

Academic Editor

PLOS ONE